# Role of plant compounds in the modulation of the conjugative transfer of pRet42a

**Luis Alfredo Bañuelos-Vazquez**[1], **Lucas G. Castellani**[2], **Abril Luchetti**[2], **David Romero**[1], **Gonzalo A. Torres Tejerizo**[2]*, **Susana Brom**[1]*

1 Programa de Ingeniería Genómica, Centro de Ciencias Genómicas, Universidad Nacional Autónoma de México, Cuernavaca, Morelos, México, 2 Departamento de Ciencias Biológicas, Facultad de Ciencias Exactas, Instituto de Biotecnología y Biología Molecular, CCT-La Plata-CONICET, Universidad Nacional de La Plata, La Plata, Argentina

\* sbrom@ccg.unam.mx (SB); gatt@biol.unlp.edu.ar (GATT)

## Abstract

One of the most studied mechanisms involved in bacterial evolution and diversification is conjugative transfer (CT) of plasmids. Plasmids able to transfer by CT often encode beneficial traits for bacterial survival under specific environmental conditions. *Rhizobium etli* CFN42 is a Gram-negative bacterium of agricultural relevance due to its symbiotic association with *Phaseolus vulgaris* through the formation of Nitrogen-fixing nodules. The genome of *R. etli* CFN42 consists of one chromosome and six large plasmids. Among these, pRet42a has been identified as a conjugative plasmid. The expression of the transfer genes is regulated by a *quorum sensing* (QS) system that includes a *traI* gene, which encodes an acyl-homoserine lactone (AHL) synthase and two transcriptional regulators (TraR and CinR). Recently, we have shown that pRet42a can perform CT on the root surface and inside nodules. The aim of this work was to determine the role of plant-related compounds in the CT of pRet42a. We found that bean root exudates or root and nodule extracts induce the CT of pRet42a in the plant rhizosphere. One possibility is that these compounds are used as nutrients, allowing the bacteria to increase their growth rate and reach the population density leading to the activation of the QS system in a shorter time. We tested if *P. vulgaris* compounds could substitute the bacterial AHL synthesized by TraI, to activate the conjugation machinery. The results showed that the transfer of pRet42a in the presence of the plant is dependent on the bacterial QS system, which cannot be substituted by plant compounds. Additionally, individual compounds of the plant exudates were evaluated; among these, some increased and others decreased the CT. With these results, we suggest that the plant could participate at different levels to modulate the CT, and that some compounds could be activating genes in the conjugation machinery.

## Introduction

Plasmid **c**onjugative **t**ransfer (CT) is a widely distributed mechanism that allows the distribution of genetic material across microorganisms. Conjugation is considered as the main

**Data Availability Statement:** All relevant data are within the manuscript.

**Funding:** This work was supported by grant IN212920 from PAPIIT, DGAPA, UNAM to SB and

grants PICT2016-0210 and PIP 2014-0420 to GTT.
LABV received a Fellowship 384814 from Consejo
Nacional de Ciencia y Tecnología. There was no
additional external funding received for this study.
The funders had no role in study design, data
collection and analysis, decision to publish, or
preparation of the manuscript.

**Competing interests:** The authors have declared
that no competing interests exist.

mechanism of DNA exchange between bacteria [1], which may belong to the same or different species. CT even occurs between organisms from different kingdoms [2]. The molecular mechanism of CT has been described and usually involves several proteins that participate in DNA processing and in the assembly of the conjugative pore; consequently, activation of CT results in an expensive metabolic burden that has to be tightly regulated [3]. Different mechanisms have evolved to regulate the synthesis of the proteins that are needed for CT. Molecules from the environment or synthetized by diverse organisms have been pointed out as regulators of the CT machinery: antibiotics [4, 5], peptides [6], plant molecules [7–10], metals [11–13] and acyl-homoserine lactones (AHLs) [7] act directly or indirectly with transcriptional activators/repressors.

Rhizobia are bacteria that grow in soils and interact with plants. Many of them have properties that improve the plant growth, *i.e.* they are able to convert the atmospheric nitrogen into ammonia in a specific-developed organ, the nodule, generated on the plant roots. Due to this property, rhizobia are used as bioinoculants, providing a cheaper and ecological friendly procedure for plant fertilization [14]. For the nodule formation and the establishment of the symbiosis, the plant secretes flavonoids under low nitrogen conditions, which in the proper rhizobia induce the production of nodulation factors (NFs). NFs start a complex signaling pathway in the plant, that includes duplication of the cells in the legume roots and several biochemical changes that prepare the plant for the rhizobial infection and finish with the nodules formation [15]. Commonly, rhizobia harbor one or more (mega) plasmids, which differ in their sizes (from a few Kbp to Mbp) [16, 17]. In some cases, these plasmids carry determinants needed for the nitrogen fixation functions, among others. Transfer of rhizobial plasmids through conjugation has been described for different species belonging to the *Rhizobiaceae*. In 2009, Ding et al. [18] classified the plasmids of rhizobia into different groups; those that are regulated by quorum-sensing (QS) -Group I-, by the *rctA-rctB* genes -Group II- and a third and fourth groups were proposed, whose regulation is still fuzzy -Groups III and IV- [19–21]. This classification was based on the phylogenetic analysis of the relaxase (*traA*). The plasmids regulated by QS, generally have a master transcriptional regulator, TraR (LuxR family regulators), which usually binds a specific AHL to activate the transcription of the genes needed for CT. These genes process and prepare the DNA for transfer (DTR) or encode components for the mating pair formation (MPF) structure. It has been recently described that in the plasmids belonging to Group I, regulated by QS, the phylogenetic analysis of TraR shows four different subgroups [22]. Group I-A is mainly composed by *Agrobacterium* plasmids, where plant signals and AHLs are needed for the activation of transfer [23, 24]. The regulation of most plasmids from group I-B is based on AHLs [25, 26]. The regulation of plasmids from group I-C does not rely on QS, nevertheless, it depends on a TraR regulator and on the genomic background [22, 27]. Also, plasmid pRL8JI from group I-C was described to respond to homoserine, a compound found in peas exudates [10]. There are no evidences yet about regulation of plasmids belonging to group I-D.

Integrative Conjugative Elements (ICEs), similarly to plasmids, can be transferred between bacteria [3]. ICEs from rhizobia, also known as Symbiotic Islands (SI), usually harbor genes needed for nodule formation and nitrogen fixation [9, 28]. ICE*Ml*sym^R7A from *Mesorhizobium loti* and ICE^Ac from *Azorhizobium caulinodans* are the most studied SIs [9, 28]. These SIs can be transferred and, remarkably, their transfer mechanism can be regulated by QS [29] or by plant compounds [9]. The transfer of ICE*Ml*sym^R7A relies on a QS mechanism that involves a LuxRI system and hypothetical genes that regulate the excision and transfer of the element [30]. It was recently demonstrated that the flavanone naringenin enhances the conjugative transfer of the SI of *A. caulinodans* in the rhizosphere of *Sesbania rostrata*, through a mechanism that involves a *lysR*-like gene, homologous to one of the master regulators of the

induction of NFs production in the rhizobia, *nodD* [9]. Plasmid pRet42a, from *Rhizobium etli* CFN42, harbors a complete DTR and MPF region of genes and a QS-regulation, dependent on AHLs [25]. Furthermore, new regulators of CT were identified on this plasmid [31]. Recently, it has been described that plasmid pRet42a is transferred at high frequencies inside the nodules developed in *Phaseolus vulgaris* (common bean), and on the root surface [32]. These results open the question of whether other molecules from the plant or the rhizosphere could modulate the transfer of pRet42a.

## Materials and methods

### Bacterial strains and growth conditions

Strains were grown on PY (0.5% peptone, 0.3% yeast extract, and 0.07 M $CaCl_2$) and in minimal medium (MM, 1.26 mM $K_2HPO_4$, 0.83 mM $MgSO_4$, 0.0184 mM $FeCl_3 \cdot 6H_2O$ and 1.49 mM $CaCl_2$ supplemented with a carbon, succinic acid 0.01 M, and nitrogen source, ammonium chloride 0.01M) at 30 ºC [33, 34]. Antibiotics were added at the following concentrations (in µg/ml): Nalidixic Acid (Nal) 20, Gentamicin (Gm) 30, Spectinomycin (Sp) 100, Rifampicin (Rif) 50 (Table 1).

### Flavonoids, exudates and extract compounds

Flavonoids: (±)-Naringenin ≥95% ((±)-2,3-Dihydro-5,7-dihydroxy-2-(4hydroxyphenyl)-4*H*-1-benzopyran-4-one, 4′,5,7-Trihydroxyflavanone), Genistein ≥98% (4′,5,7-Trihydroxyisoflavone, 5,7-Dihydroxy-3-(4-hydroxyphenyl)-4H-1-benzopyran-4-one), Quercetin ≥98% (2-(3,4-Dihydroxyphenyl)-3,5,7-trihydroxy-4*H*-1-benzopyran-4-one dihydrate, 3,3′,4′,5,7-Pentahydroxyflavone dihydrate), Acetosyringone (4′-Hydroxy-3′,5′-dimethoxyacetophenone), Luteolin ≥98% (3′,4′,5,7-Tetrahydroxyflavone), Apigenin ≥95.0% (4′,5,7-Trihydroxyflavone, 5,7-Dihydroxy-2-(4-hydroxyphenyl)-4-benzopyrone), and Gallic acid 97.5–102.5% (titration) (3,4,5-Trihydroxybenzoic acid) were purchased from Sigma-Aldrich Chemicals (St. Louis, MO). Stock solutions were prepared at 100 mM in a 1:1 mixture of water and DMSO.

For the preparation of exudates of *P. vulgaris* and *M. sativa*, 20 plants were placed in 500 ml of nitrogen-free Fåhraeus nutrient solution [38] for 5 days post germination (dpg). Afterwards, the medium was recovered, filtered through a 0.22 µm filter and stored at 4 ºC in the dark.

For the preparation of roots extracts of *P. vulgaris*, three of the plants used to obtain exudates were used. The roots of these plants were macerated in 20 ml of nitrogen-free Fåhraeus nutrient solution, filtered through a 0.22 µm filter, and maintained at 4 ºC in the dark.

**Table 1. Strains and plasmids used in this work.**

| Strains and plasmids | Relevant characteristics | Features | References |
|---|---|---|---|
| ***R. etli*** | | | |
| CFNX182 | Derivative of *R. etli* CFN42 cured of pRet42a | Nal | [35] |
| CFNX559 | Derivative of CFNX182 with RFP in the chromosome and GFP in pRet42a | Nal, Gm, Sp, RFP and GFP | [36] |
| CFN2001 | CFN42 derivative (pRet42a⁻ pRet42d⁻) | Rif | [37] |
| CFNX187 | CFNX182 complemented with pRet42a::Tn5*mob* | Km, Nm | [35] |
| CFNX669 | CFN42 derivative, *traI*::pSUPΩSp | Sp | [25] |

Nal, Rif, Nm, Sp, Gm, Km = nalidixic acid, rifampicin, neomycin, spectinomycin, gentamicin and kanamycin resistance, respectively. RFP, red fluorescent protein. GFP, green fluorescent protein

For the nodules extracts, 150 nodules were collected from 21 days post inoculation (dpi), macerated in 20 ml of nitrogen-free Fåhraeus nutrient solution, filtered with a 0.22 μm filter and maintained at 4 ºC in the dark.

## Bacterial matings

Conjugations between the strains were done biparentally, using overnight cultures grown to stationary phase, *ca.* 1.2 $OD_{600nm}$ [25]. Conjugations in liquid medium were done as follows: donor and recipient strains were mixed in a 1:1 volume ratio in 5 ml of PY or MM medium at a final $OD_{600nm}$ of 0.05 and incubated at 30 ºC overnight at 200 rpm. Flavonoids were included, respectively, at different concentrations (2 μM, 20 μM, and 50 μM). Conjugations with exudates and extracts of roots and nodules were done in liquid medium by mixing 2.5 ml of Fåhraeus containing the exudates or extracts (or nitrogen-free Fåhraeus nutrient solution as control), 2.5 ml of PY medium, 0.25 ml of donor and 0.25 ml recipient (each *ca.* 1.2 $OD_{600nm}$). The mixtures were incubated at 30 ºC overnight, and then centrifuged at 5000 rpm for 2 minutes and the supernatant was discarded. The pellet was suspended in 1 ml of 10 mM $MgSO_4$ 0.01% Tween 40 (vol/vol). Conjugations in an environment of low oxygen were done by mixing the donor and recipient strains in a 1:1 volume ratio on solid medium and placed in a chamber under a 1% oxygen / 99% argon mixture and incubated at 30 ºC overnight. As control, a 1: 1 mixture of water and DMSO was made and the same amount was added to the conjugation assays for the different concentrations used for the different compounds. Serial dilutions were plated on the selective media supplemented with the corresponding antibiotics, to quantify the donor, recipient and transconjugants cells. The conjugation frequency is expressed as the number of transconjugants per donor cell.

## Plant assays

Seeds from *P. vulgaris* cv Negro Jamapa were sterilized and germinated as previously described by Bañuelos-Vazquez et al [32]. Seedlings of two dpg (3–4 cm in length) were introduced in tubes with 40 ml of nitrogen-free Fåhraeus nutrient solution [38] and in the indicated experiments, also supplemented with a carbon (succinic acid 0.01M) and nitrogen source (ammonium chloride 0.01M) [38]. Tubes were inoculated with donor and recipient strains adjusted at a final 0.05 $OD_{600nm}$, in a 1:1 ratio. The conjugation frequency was measured at 1, 10 and 20 dpi. For each experiment, the medium of three tubes (in presence or absence of plants) was collected. The medium from the tubes were centrifugated at 5000 rpm, supernatant was discarded, and the pellet was resuspended in 1 ml of 10 mM $MgSO_4$ 0.01% Tween 40 (vol/vol). Serial dilutions were plated on the selective media supplemented with the corresponding antibiotics. Root samples were introduced in Falcon tubes with 30 ml of nitrogen-free Fåhraeus nutrient solution and subjected to ultrasound for 20 min in a Branson 200 ultrasonic cleaner. Then, the roots were taken out and the medium was centrifuged for 15 min at 5000 rpm, at 4 ºC to recover the bacteria removed from the root's surface. The pellet was resuspended in 1 ml of 10 mM $MgSO_4$ 0.01% Tween 40 (vol/vol). Serial dilutions were plated on the selective media supplemented with the corresponding antibiotics. All experiments were repeated at least three times.

## Statistical analysis

Statistical analyses were performed using GraphPad Prism (version 8.0) software (GraphPad Software, La Jolla, CA). Statistical significance was determined using two-tailed unpaired Student's *t*-test or one-way ANOVA with Dunnett´s multiple comparisons test. A *p*-value $\leq$ 0.05 was considered significant.

## Results

### Conjugative transfer of pRet42a in the presence of *Phaseolus vulgaris, Zea mays* and *Medicago sativa* plants

In a previous work, we showed the capacity of plasmid pRet42a of *R. etli* CFN42 (*i.e* derivative CFNX559) to be transferred at high frequency to other bacteria on the surface of the roots and inside the nodules generated during the interaction between this strain and *P. vulgaris* [32]. This phenomenon led us to consider that a set of plant molecules could be involved in the modulation of CT of pRet42a. To address this question, we performed CT experiments in presence and absence of bean plants (*P. vulgaris*). Seedlings of two days post germination (dpg) were introduced in tubes with nitrogen- free Fåhraeus nutrient solution [38]. Tubes were inoculated with CFNX559 as a donor and CFN2001 as a recipient strain, and the CT frequencies were evaluated in the medium of the tubes (with or without plant), and on the root surface. CFNX559, the donor strain, has the red fluorescent protein marker (RFP) inserted into the chromosome and the green fluorescent protein marker (GFP) in pRet42a. Therefore, these cells have both fluorescences. When transferring the plasmid to the recipient cells that do not have any fluorescent marker, the transconjugants only express the GFP, allowing differential detection. The frequencies were evaluated at 1, 10- and 20-days-post-inoculation (dpi). The values of CT frequency obtained for the media with and without plants were very similar at 1 or 10 dpi (Fig 1A). However, there were no transconjugants at 20 dpi in absence of plants, while there were CT events in the media from tubes containing plants. It is worth to mention that when comparing the number of donor and recipient bacteria at 10 dpi and 20 dpi, there

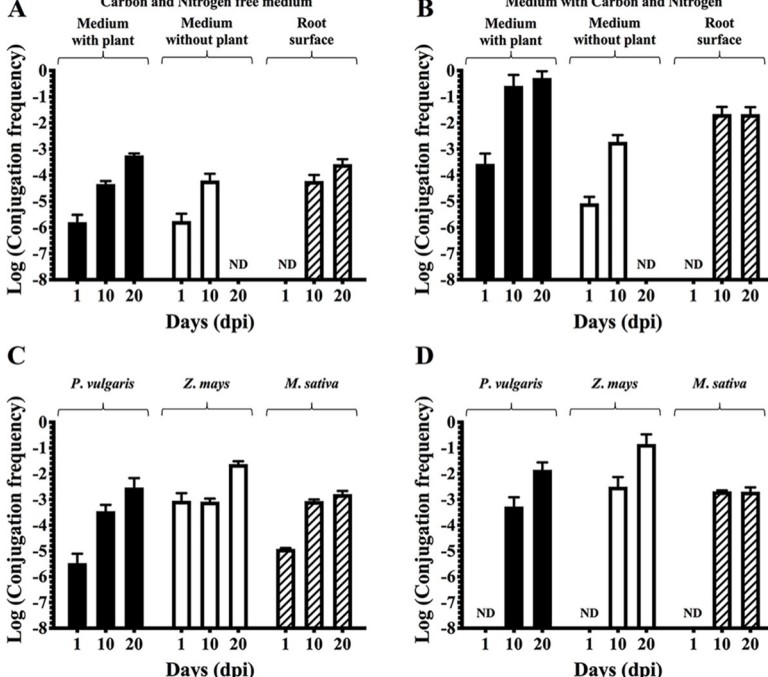

**Fig 1. CT frequency in response to presence of different plants and conditions. A.** Evaluation of CT frequencies of pRet42a in presence or absence of *P. vulgaris* and on the root surface, in media without C and N sources. **B.** Evaluation of CT frequencies of pRet42a in presence or absence of *P. vulgaris*, and on the root surface, in media supplemented with C and N. **C.** Evaluation of CT frequencies of pRet42a in media, in presence of *P. vulgaris*, *Z. mays* and *M. sativa*. **D.** Evaluation of CT frequencies of pRet42a on the roots of *P. vulgaris*, *Z. mays* and *M. sativa*. All the experiments were performed at 1, 10- and 20-days post-inoculation (dpi). ND, not detected.

were *ca.* 2–4 million less CFU at 20 dpi. On the root surface, the CT frequencies at 10 and 20 dpi were relatively high and similar to that of media with plants. Nevertheless, CT was not observed in the rhizoplane at one dpi. A possible explanation is that 24 hours is not enough time for the bacteria to reach the root, attach and conjugate. The lack of transconjugants at 20 dpi and the decrease in donor and recipient bacteria in the plant-free media allows us to consider the possibility that this may be caused by a nutrient limitation in the absence of plants. Therefore, a similar experiment was made with the medium supplemented with N and C sources (Fig 1B). Addition of the N and C sources led to an increase of the CT in presence of bean plants at all time-points. Also, the CT frequency in the absence of plants at 1 and 10 dpi increased, but CT could still not be observed at 20 dpi. Moreover, we compared the conjugation frequency with two other plants: *Zea mays* (non-legume) and *Medicago sativa* (non-host-legume). CT evaluated in the medium where the plants were placed (Fig 1C) showed that, at one dpi the CT frequency in *Z. mays* was higher than in *P. vulgaris* and *M. sativa*, but at 10 and 20 days the CT was very similar among the three plants. This suggests that compounds released by *Zea mays* at the early times after inoculation may be more abundant, resulting in a richer medium for bacteria to reproduce and reach the QS threshold for conjugation in a shorter lapse. In addition, the bacteria that adhered to the root surface were analyzed. The results showed that there was no significant difference at 10 and 20 dpi, while CT was not observed at one dpi (Fig 1D). All these results taken together suggest that plants play an important role in the CT of pRet42a.

## Conjugative transfer of pRet42a in response to different types of molecules

Exudates from leguminous plants contain molecules involved in the first steps of the symbiotic process, and the presence of some of these molecules often defines the specificity of the symbiosis [39]. Considering that the presence of bean roots in the media enhances CT of pRet42a, we asked ourselves if a specific compound from the plant, or a mixture of them, could be responsible for the CT modulation. Firstly, the CT frequency of pRet42a was evaluated in presence of *P. vulgaris* exudates. The presence of *P. vulgaris* exudates induced very high values of conjugative frequency in comparison with those obtained for the media without exudates (Fig 2A). To evaluate if this effect was plant-specific, experiments were performed using exudates from *M. sativa*, which is not a host of *R. etli*. Increased values were obtained in comparison with the control, but the CT frequencies were lower than those obtained in presence of *P. vulgaris* exudates. These results indicate that molecules present in both exudates act as conjugative

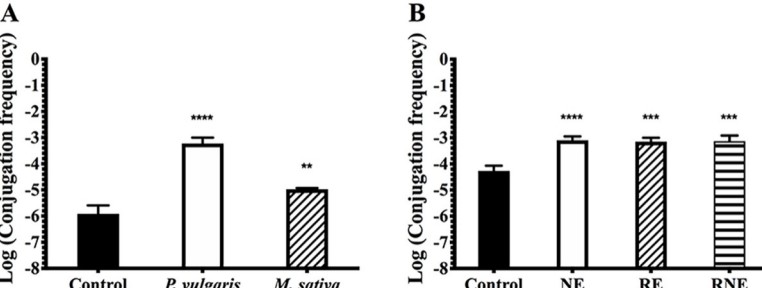

**Fig 2. CT frequency in response to plant exudates and extracts. A.** Evaluation of CT frequencies of pRet42a in response to root exudates from *P. vulgaris* and *M. sativa*. **B.** Evaluation of CT frequencies of pRet42a in response to extracts of root and nodules of *P. vulgaris* plants. Statistical analysis was performed by one-way ANOVA (*p value* < 0.05) with Dunnett´s multiple comparisons test. Key: ** (*p value* = 0.0088), *** (*p value* = 0.0001). **** (*p value* < 0.0001). For statistical analyses, each treatment was compared to the control condition.

inducers. As the composition of exudates differs from one plant to another [40], probably some specific molecules present in *P. vulgaris*, but not in *M. sativa* exudates, are responsible for improving to a greater extent the CT of pRet42a.

Previous work showed that CT frequencies are relatively high inside the nodules [32, 36]. Moreover, high CT frequencies were also observed on the root surface. Thus, extracts from nodules and roots of *P. vulgaris* were prepared and evaluated as inducers of CT. The presence of nodule or root extracts resulted in an increase in CT frequencies (Fig 2B). The effect of the mixture of both extracts (root and nodules) was similar to that of each extract individually.

## Conjugative transfer of pRet42a in response to flavonoids

Flavonoids are a group of molecules present in plant exudates [41]. The flavonoid composition of exudates is specific for each plant, and is involved in the first steps of the interaction with the symbiont [39, 41] Table 2 shows the main characteristics of the plant compounds used in this work. To determine if the enhancement of CT in presence of the plant is related to the flavonoids secreted by the plant, we evaluated their role in CT of pRet42a. Conjugation assays were performed in presence of three different concentrations of seven flavonoids: naringenin, genistein, quercetin, acetosyringone, luteolin, apigenin and gallic acid. At a 2 μM concentration, two compounds enhanced CT, naringenin and apigenin (Fig 3A). At concentrations of 20 μM, none of the flavonoids showed significant differences compared to the control, although naringenin showed a tendency to higher values than the other molecules (Fig 3B). At a concentration of 50 μM, naringenin generated statistically significant higher values than the control (showing a similar behavior at the three different concentrations), while quercetin showed a decrease in CT frequency values (Fig 3C). It should be noted that a mixture of the seven flavonoids did not show differences with control values, at any concentration, indicating that a specific proportion of the different molecules is needed. Keeping in mind that the response to a specific flavonoid depends on the concentration of each molecule, the transfer of pRet42a in response to apigenin was evaluated at different concentrations between 0.5–20 μM

**Table 2. Characteristics of the different plant compounds used.**

| Name | polyphenol class | polyphenol Sub-class | Characteristics | References |
|------|------------------|----------------------|-----------------|------------|
| Naringenin | Flavonoid | Flavanone | • Induces the transfer of an 87.6-kb integrative and conjugative element that is excised and transferred by CT in *Sesbania rostrata*. | [9, 40, 42, 43] |
| | | | • Reduces production of acyl homoserine lactones (AHLs) in *Pseudomonas aeruginosa* PA01. | |
| | | | • Nodulation factor inductor of *Rhizobium leguminosarum* biovar *phaseoli*. | |
| | | | • Produced by some ecotypes of *Phaseolus vulgaris*, *Vicia sativa* and soybeans. | |
| Genistein | Flavonoid | Isoflavonoid | • Nodulation factor inductor of *Bradyrhizobium japonicum*. | [40, 44, 45] |
| | | | • Antifungal and antibacterial activities. | |
| | | | • Produced by some ecotypes of *P. vulgaris* and soybeans. | |
| Quercetin | Flavonoid | Flavonols | • Antibacterial activity. | [46, 47] |
| | | | • Produced by some ecotypes of *P. vulgaris*. | |
| Luteolin | Flavonoid | Flavones | • Nodulation factor inductor of *Ensifer melilloti*. | [48, 49] |
| | | | • Produced by *Medicago* spp. *(e.g. M. sativa)* | |
| Apigenin | Flavonoid | Isoflavonoid | • Nodulation factor inductor of *R. leguminosarum* biovar *phaseoli*. | [40, 50–52] |
| | | | • Produced by peas and soybean. | |
| Gallic acid | Phenolic acids | Hydroxybenzoic acids | • Produced by some ecotypes of *P. vulgaris*. | [53] |
| Acetosyringone | Phenolic compounds | | • Inductor of virulence genes in *Agrobacterium tumefaciens*. | [54, 55] |
| | | | • Produced by *Nicotiana tabacum* and many other dicotyledonous plants. | |

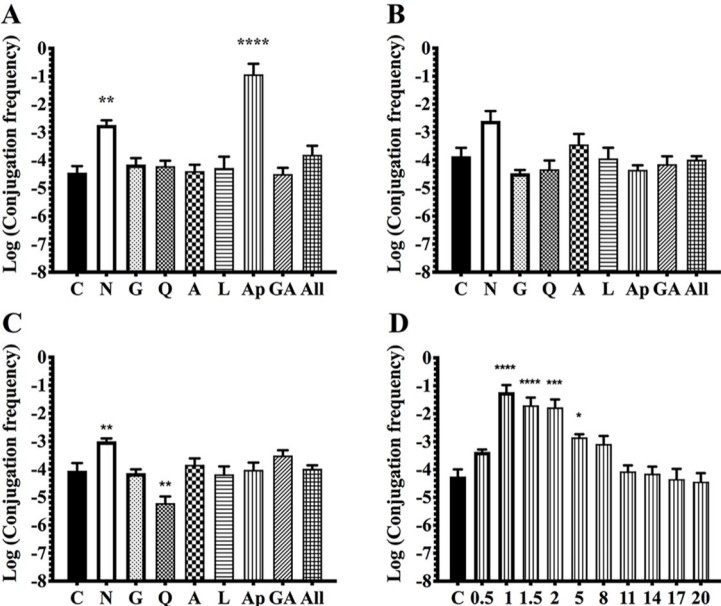

**Fig 3. CT frequency in response to different flavonoids.** Evaluation of CT frequencies of pRet42a in response to naringenin (N), genistein (G), quercetin (Q), acetosyringone (A), luteolin (L), apigenin (Ap), gallic acid (AG) and all the flavonoids together (All), at 2 μM (**A**), 20 μM (**B**) and 50 μM (**C**). **D**. Evaluation of conjugation frequencies of pRet42a in response to different concentrations of apigenin. Statistical analysis was performed by one-way ANOVA ($p$ $value < 0.05$) with Dunnett´s multiple comparisons test. Key: In **A**, ** ($p$ $value = 0.0047$), **** ($p$ $value < 0.0001$). In **C**, ** ($p$ $value < 0.003$). In **D**, * ($p$ $value = 0.0135$), ** ($p$ $value = 0.0002$), **** ($p$ $value < 0.0001$). For statistical analyses, each treatment was compared to the control condition (C).

(Fig 3D). Concentrations from 1–2 μM of apigenin induced the highest CT increase (Fig 3D). As naringenin showed an enhanced CT frequency at 2 μM, it was also evaluated in a variable range of concentrations (0.5, 1, 2, 5, 11 and 20 μM) but CT frequencies did not vary in comparison to 2 μM (not shown).

## pRet42a transfer is QS dependent in presence of bean plants

As mentioned before, transfer of pRet42a is regulated by a QS system, characterized by the interaction between the TraR regulator and the AHLs produced by TraI [25]. Aiming to analyze if the CT induction in presence of plants relies on the QS system of pRet42a, a *traI* mutant (CFNX669) was evaluated. Thus, CT frequencies of CFNX669 in presence of bean plants were assayed at different times after inoculation. Fig 4 shows that in presence of the complete QS system (strain CFNX187), the plasmid was transferred, both in presence of the plant and on the root surface, similar to the results obtained with strain CFNX559. Nevertheless, the *traI* mutant was not able to conjugate in the medium nor on the root surface of the plant. These results indicate that although some molecules present in plant exudates improve CT of pRet42a, the QS system is essential for plasmid transfer.

## pRet42a transfer responds to environmental conditions

Even though we have shown that pRet42a CT responds to molecules present in plant exudates and in nodules or root extracts, it is also possible that other environmental conditions could be involved in the modulation of CT. Specifically, three conditions of the rhizosphere or inside the nodules were evaluated. First, experiments were performed to evaluate plasmid transfer in

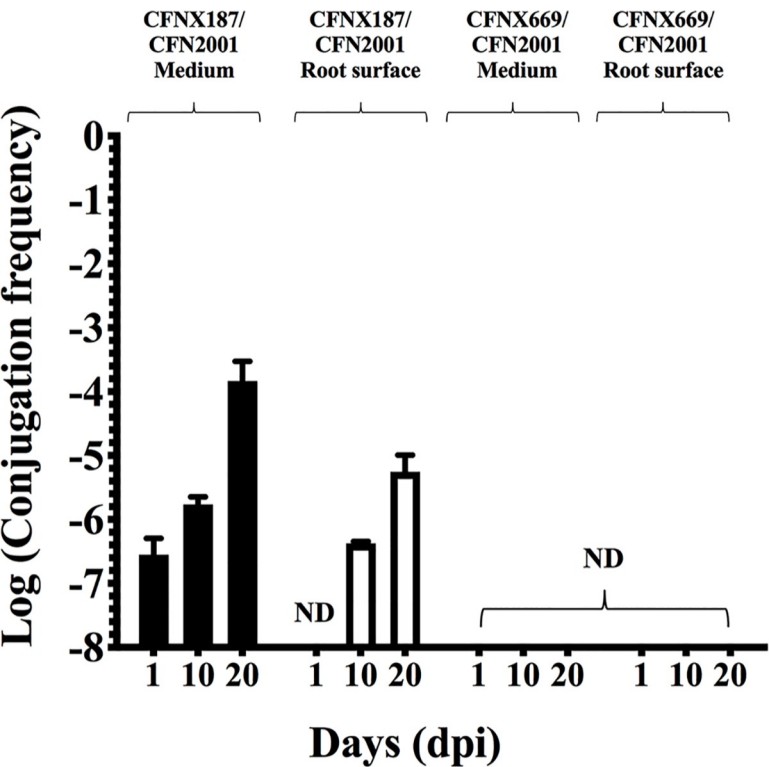

**Fig 4. CT frequency of a QS regulatory mutant, in the media in presence of *P. vulgaris* plants and on the root surface.** Evaluation of CT frequencies of pRet42a (CFNX187) and a *traI* mutant of this plasmid (CFNX669) in the media in presence or absence of *P. vulgaris*, and on the root surface at 1, 10 and 20 dpi. ND, not detected.

response to a reduced oxygen condition, similar to that observed inside the nodule. We obtained higher conjugation frequency values in a low $O_2$ environment (see materials and methods) (Fig 5A), suggesting that this condition inside the nodule may be partially responsible for the CT induction observed in our previous work [32]. In addition, we compared plasmid transfer between two different media; minimal and enriched media. We found that in an enriched media, conjugation is remarkably higher than in minimal media, suggesting that the nutrients provided by the plant to the nodule could also contribute to enhance conjugation in

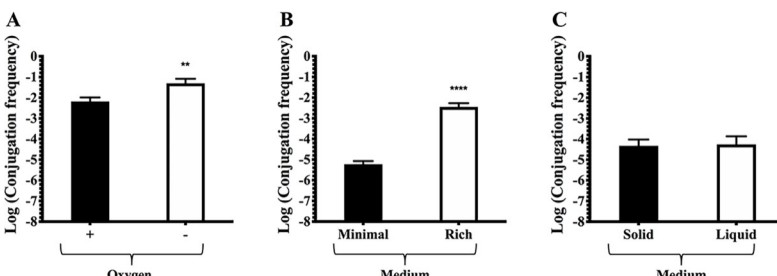

**Fig 5. pRet42a CT frequency in response to different environmental conditions. A.** Different oxygen concentrations. **B.** Nutrient composition of the media. **C.** Solid or liquid media. Statistical comparison was performed through a *t test* (*p value* < 0.05) in comparison to the control condition. Key: ** (*p value* = 0.0015), **** (*p value* < 0.0001).

the rhizosphere and inside the nodules (Fig 5B). Finally, we tested if the mating was different for *R. etli* in liquid or over a surface, but no differences were observed (Fig 5C).

## Discussion

Plasmids are vastly distributed in bacteria and CT is the main mechanism by which they are transferred. CT requires a set of proteins that have to recognize the origin of transfer, make a cleavage on it and couple the DNA strand to the system that will transfer it, mainly through a T4SS, which is also known as MPF. During the whole process, several proteins are needed, thus, it is an energetically expensive process. To avoid draining the cellular energy, several mechanisms of CT regulation have been found. In rhizobia, there are mainly two mechanisms, (QS and *rctA/B*, see introduction for details; [56]), but recently, molecules from the environment have been described as inductors of CT. Flavonoids produced by *Sesbania rostrata*, induce the transfer of an 87.6-kb ICE [9]. Moreover, we have described that plasmid pRet42a from *R. etli* CFN42 is transferred at high rates inside the nodules developed by *R. etli* in *P. vulgaris* and on the root surface [32, 36]. With the goal of extending the knowledge, and understanding the high rates of CT observed during the symbiotic process, we characterized the response of the CT of pRet42a to compounds produced by different plants and to some environmental conditions.

Our first strategy was to perform conjugations in presence and absence of distinct plants (*P. vulgaris*, *M. sativa* and *Zea mays*) at different times. This strategy is also supported by previous works, which described that homoserine, a compound secreted by pea plants induces CT of pRL8JI [10]. Our results showed that the sole presence of the plant plays an important role in increasing the conjugation frequency. This phenomenon could be due to secretion of signals that can be involved in regulation of CT, secretion of metabolites that the bacteria could use as energy supplies or as a support where the bacteria can adhere and conjugate. Moreover, the lack of transconjugants (and the reduction of donor and recipient bacteria) at 20 dpi in absence of plants was remarkable (Fig 1). This result suggests that plants could also be important for long-term bacterial survival. Plants secrete a wide spectrum of molecules, including sugars, aminoacids, phenolic compounds, lipids and even nucleotides [57, 58]. All these molecules could directly enhance CT or also be used as nutrients, supporting the hypothesis that they may be involved in survival of the bacteria.

Following this research line, we evaluated if the exudates or the compounds secreted by the plant affected the CT. The results indicate that exudates from the native host plant, *P. vulgaris*, and also from non-native host plants as *M. sativa* and *Z. mays*, can act as conjugative transfer inducers (Fig 2). It has been described that the composition of exudates differs from one plant to another [40]. Thus, the CT could be changing according to the composition and quantity of the molecules present in the different exudates. Another possibility is that differences in the proportion of molecules in the two exudates are responsible for the levels of CT induction, since certain specific compounds can increase CT while others limit it. Nodule and root extracts were also evaluated as inducers (Fig 2). The values obtained were also above the control. Thus, some molecules present both in roots and nodules may modulate the CT. Based on the non-additive effect of both fractions, the molecules involved could be very similar or act through the same molecular mechanism. This induction could explain the high frequency obtained inside the nodules [32].

Flavonoids are key molecules in rhizobia-legume symbiosis. The specificity of their response determines the outcome of the symbiotic relationship [39, 41]. The bacteria induce the expression of *nod* genes in response to some flavonoids. For these reasons, the role of flavonoids was evaluated (Fig 3). Naringenin and apigenin showed induction of CT while quercetin

showed, at high concentrations, a negative effect on CT rates. Apigenin shows a higher induction of CT when present in concentrations of 1–2 μM, while induction with naringenin seems to be constant at different concentrations. It has been described that the transfer of the SIs of *Azorhizobium caulinodans* is induced by plant-secreted flavonoids through AhaR, a LysR-family protein homolog of NodD [9]. In *Rhizobium tropici* CIAT 899, an AraC-type regulator protein named OnfD was recently described [59]. OnfD is involved in the synthesis of NFs and may form heterodimers with NodD, thus playing a role in the transcriptional activation of the *nod* genes. A possible explanation for our results is that, in *R. etli* CFN42, flavonoids (or other plants compounds) interact either with an unknown regulator from the LysR-family (*R. etli* CFN42 has 69 annotated proteins from the LysR-family of proteins) or with NodD (*R. etli* CFN42 has three copies of NodD), to form a complex, which may modulate (indirectly or directly) the expression of genes involved in CT. Further experiments are needed to resolve which are the genes and targets involved (Fig 6). To gain insight about the genes and molecules involved in this signaling, two strategies could be performed. In a first approach, induction of CT by plant compounds could be studied in plasmid-cured derivatives of CFN42, to determine if genes present in these replicons are needed for induction of CT, similar to the strategy performed by Yost et al [60]. Another alternative could be to use transcriptional fusions of the regulatory regions of *tra* genes and evaluate, in an extensive way, the effect of other compounds [61]. It cannot be disregarded that regulators with XRE-domains (helix-turn-helix domain) could be involved in the detection of plant molecules. XRE regulators have been described in pRet42a of *R. etli* CFN42 as RHE_PA00165 [31], in pRleVF39b of *R. leguminosarum* VF39SM as *trbR* [19] and in ICE*Ml*SymR7A of *M. loti* as *qseC* [62]. RHE_PA00165 is needed by pRet42a for its transfer from genomic backgrounds different than the wild-type. In *R. leguminosarum* VF39SM, a mutation in *trbR* leads to an increase in pRleVF39b CT. Meanwhile, *qseC* modulates the excision and conjugative transfer of ICEMlSym$^{R7A}$.

QS signal mimics produced by plants have been described to interfere with the QS communication between bacteria; this appears to be particularly prevalent among nodulating plants [63]. A mutation in *traI* abolishes transfer of pRet42a under laboratory conditions [25]. To find out if some of the flavonoids or other compound from the plant could be acting through the QS system, we evaluated if the presence of the plant could replace the AHL of pRet42a. The results showed that the *traI* mutant was unable to transfer in presence of the plant, indicating that plant secreted compounds could not substitute the bacterial AHL for CT activation (Fig 4). Even if plant compounds could form a complex to modulate CT, the regulatory circuit between AHLs and TraR is indispensable (Fig 6).

Finally, the effect of some environmental conditions was tested (Fig 5). Usually, CT of rhizobial plasmids is evaluated on plates with solid medium. Nevertheless, some reports show that a non-rhizobial plasmid (RP4) is transferred from a rhizobia strain in liquid medium [64, 65]. We observed that pRet42a was able to conjugate well in liquid media (Fig 5C). To our knowledge, there is no other published evidence of conjugation of rhizobial plasmids in liquid media. These findings may imply that the Mpf genes of pRet42a encode a flexible pilus [66, 67]. The CT rates in rich media were higher than in minimal media, suggesting that nutrient availability, and thus cellular energy content, positively affects conjugation. In a low $O_2$ environment (emulating the conditions inside the nodule), CT frequencies were slightly higher. These results could explain that this condition inside the nodule may be partially responsible for the plasmid behavior observed in our previous work [32], where the CT was high inside the nodules.

Transfer of plasmids via conjugation and the mechanisms involved in the molecular regulation of this phenomenon are still under study. Here, the role of environmental conditions and diverse compounds produced by the plant were evaluated. These results showed that CT may

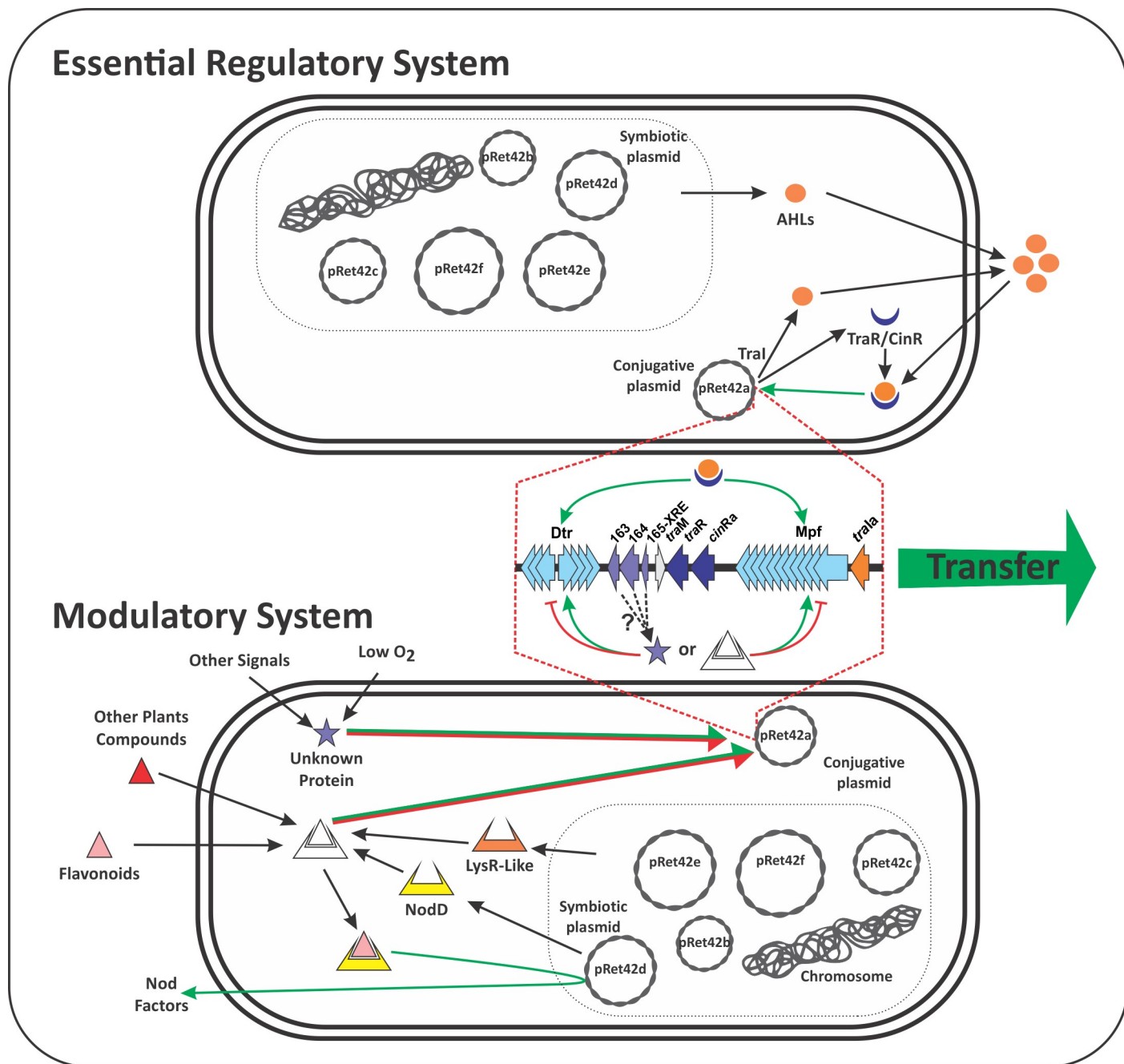

**Fig 6. Scheme of possible pathways involved in pRet42a CT modulation.** The essential regulatory system (Top) figure shows the effect of the AHLs (orange circles) produced by the different replicons of *R. etli* CFN42 on the activation of pRet42a transfer. Despite that transfer responds to different AHLs, the interaction between AHLs produced by TraI and TraR/CinR regulators is essential for pRet42a transfer. In the Modulatory System (Bottom), we propose a mechanism for the effect of flavonoids, other plant compounds and environmental signals as low oxygen conditions. For flavonoids and other plant compounds (triangles), we propose an interaction with pRet42d encoded NodD regulators, or with other LysR-like regulators present in other replicons. Since NodD-flavonoids triggers Nod Factors expression, a similar mechanism could be involved in the modulation of CT of pRet42a. For the other signals, as $O_2$ concentration, an unknown protein (star) could sense the signal and then modulate transfer genes expression. These "star" proteins might be the hypothetical proteins located between Dtr and Mpf genes. The dotted line indicates that the contained replicons could produce AHLs (Top) or LysR-like proteins (Bottom). Green arrows indicate activation or production; red arrows indicate repression.

be modulated by different molecules, where alternative regulatory mechanisms could be involved. Thus, the panorama gets more complex, including more (new) actors that should be analyzed in the near future.

## Acknowledgments

LGC and AL are fellows of CONICET. GTT is member of the Research Career of CONICET. For technical support, Alfonso Leija and Georgina Hernández from the Programa de Genómica Funcional de Eucariotes, CCG, UNAM, for providing *P. vulgaris* seeds. The authors wish to thank Dr. Maria Esperanza Ruiz for help with the statistical analysis.

## Author Contributions

**Conceptualization:** Luis Alfredo Bañuelos-Vazquez, David Romero, Gonzalo A. Torres Tejerizo, Susana Brom.

**Formal analysis:** David Romero, Gonzalo A. Torres Tejerizo, Susana Brom.

**Funding acquisition:** David Romero, Gonzalo A. Torres Tejerizo, Susana Brom.

**Investigation:** Luis Alfredo Bañuelos-Vazquez, Lucas G. Castellani, Abril Luchetti, David Romero, Gonzalo A. Torres Tejerizo, Susana Brom.

**Methodology:** Luis Alfredo Bañuelos-Vazquez, Lucas G. Castellani, Abril Luchetti, Gonzalo A. Torres Tejerizo, Susana Brom.

**Project administration:** Susana Brom.

**Supervision:** David Romero, Gonzalo A. Torres Tejerizo, Susana Brom.

**Writing – original draft:** Luis Alfredo Bañuelos-Vazquez, Lucas G. Castellani, Gonzalo A. Torres Tejerizo, Susana Brom.

**Writing – review & editing:** Luis Alfredo Bañuelos-Vazquez, Lucas G. Castellani, Abril Luchetti, David Romero, Gonzalo A. Torres Tejerizo, Susana Brom.

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
