## [Decision Letter · Decision Letter 0]

28 Jul 2020

PONE-D-20-19518

Role of plant compounds in the modulation of the conjugative transfer of pRet42a

PLOS ONE

Dear Dr. %Torres-Tejerizo%,

Thank you for submitting your manuscript to PLOS ONE. After careful consideration, we feel that it has merit but does not fully meet PLOS ONE’s publication criteria as it currently stands. Therefore, we invite you to submit a revised version of the manuscript that addresses the points (specially those concerning to the second reviewer) raised during the review process.

We look forward to receiving your revised manuscript.

Kind regards,

Francisco Martinez-Abarca, Ph.D.

Academic Editor

PLOS ONE

Journal Requirements:

2.Thank you for stating the following in the Acknowledgments Section of your manuscript:

[LABV received a Fellowship 384814 from Consejo Nacional de Ciencia y Tecnología.]

 [SB IN212920 PAPIIT, DGAPA, UNAM

GTT PICT2016-0210 ANPCYT

GTT PIP 1220130100420CO CONICET

The funders had no role in study design, data collection and analysis, decision to publish, or preparation of the manuscript.]

Reviewers' comments:

Reviewer's Responses to Questions

**Comments to the Author**

1. Is the manuscript technically sound, and do the data support the conclusions?

Reviewer #1: Yes

Reviewer #2: Yes

2. Has the statistical analysis been performed appropriately and rigorously? 

Reviewer #1: Yes

Reviewer #2: Yes

3. Have the authors made all data underlying the findings in their manuscript fully available?

Reviewer #1: Yes

Reviewer #2: Yes

4. Is the manuscript presented in an intelligible fashion and written in standard English?

Reviewer #1: Yes

Reviewer #2: Yes

5. Review Comments to the Author

Reviewer #1: This paper is an extension of very exciting previous work from the same group, which showed that conjugative transfer of pRetCFN42a is enhanced on plant surfaces and in nodules, using a very elegant labeling system. The current work seeks to address the nature of the effect of the plant on transfer. Different plants were examined (Corn, Alfalfa and Bean) and all promoted transfer, perhaps due to nutritional effects. Exudates from seeds/roots of alfalfa and bean were compared and there was a stronger effect of the host plant, bean. Known plant compounds were tested for stimulatory effects on conjugation, and an effect of two nodulation inducers, apigenin and naringenin, was noted. In all cases, transfer by conjugation required the AHL type QS system previously shown to be involved in transfer of this plasmid.

Overall this is a very well organized piece of work that adds to the story. It is very well written and easy to follow. I do not have too much to criticize except for minor editorial things (see below), but do have some suggestions for improving the paper or directing future work. Some of this might go in the discussion.

1) It should probably be cited that conjugation of the plasmid pRL8JI (also a type I conjugation system) is stimulated by homoserine, a non-protein amino acid that is present in high concentration in pea root exudates (Vanderlinde et al. 2014, Environ Microbiol 16: 205-217).

2) Some speculation as to other compounds that might be specific to beans or legumes in general would not be a bad idea. It is a little disappointing to look only at nod gene inducers.

3) Do the effects of naringenin and apigenin depend on the presence of a functional nodD gene ? This could be assayed fairly easily in a pSym cured derivative of CFN42, like your recipient, once it has acquired the p42a.

4) Ways of getting at what the particular compounds in exudates might be could be discussed. In this context the method used in Rosenblueth et al. (1998, MGG, 258:587) and Yost et al. (2004, Microbiology, 152:2061) could be mentioned. I.e. depletion of nutrients in exudates by plasmid cured strains to see what is left over and still has the ability to induce transfer.

5) More could be made of the fact that genes encoding Xre-type regulators have been found in conjugation systems, including the one from pRetCFN42a, strongly suggesting additional effector molecules that have an input into regulation of transfer. It has always been a strong possibility that some plant produced compounds could be the effectors that interact with these regulators.

Many experts in the field don't like the use of the word "conjugal" and much prefer conjugative. Laura Frost, whom you cite in ref 1, gets quite angry at people who use conjugal .

Line 38. Homoserine is missing an e; same for line 64 and elsewhere

Line 49. Should be a semicolon (;) not a comma, after evaluated.

Line 195. no transconjugants (not not)

Lines 301/302. Not sure "above them in hierarchical order is correct. In agrobacterium, opines come first, with QS below opines in the cascade. But QS mutants (traR or traI will still not transfer in the presence of opines. Your situation could be the same. Maybe you need to explain what you mean by hierarchy here.

Table 2. Not all of these compounds are flavonoids, so the title is inaccurate. I suggest plant compounds, or plant produced phenolic compounds.

References: some titles are written in sentence case (correct), others in title case (capital letters on most nouns and other important words) - be consistent. This is a common error caused by trusting reference manager programs and importing papers from PubMed etc. Also double check for accents on author names - some seem to be missing even on Spanish names, though this may be how the names were written on the original publication

Reviewer #2: The manuscript PONE-D-20-19518 reports data from experiments aimed to clarify the role of plant-derived compounds on the regulation of conjugative transfer of plasmid pRet42a from the legume microsymbiont Rhizobium etli. Obtained results indicate that quorum sensing system is playing the main role in the regulation, but also that compounds in plant root exudates may modulate the extent of conjugal transfer.

The results are interesting and novel, tough not conclusive over a mechanistic interpretation of the phenomenon. Consequently, the paper is mainly descriptive, though opening the way to several future experiments. A key experiment for a more mechanistic report could be the used of a nodD gene mutant strain (see point 6 below).

However, there are many points which need to be clarified and additional details reported to allow readers to better understand the results and let reproduce the experiments shown. Possible simple additional experiments could also be performed (see below).

1. Line 144. Please indicate the OD600 or the number of cells used.

2. Line 146. Concentrations of used flavonoids should be indicated here.

3. Lines 160 and followings. The plant assay must be described in detail. For instance, the volume used and the number of plants and root lengths. The number of planted cells (did you perform dilutions? Which volume of plant medium was taken?)

4. Line 195-205. Why at at 20 dpi there were not transconjugants? This is really surprising since you observe transconjugants at 10 dpi. Were rhizobial cells titres comparable between 10 dpi and 20 dpi? Or rhizobial cells died after 10 dpi? This point strongly needs a clarification and hypotheses driven by possible additional control experiments, to support your sentence on line 352.

5. Line 232. Experiment with plant root exudates must be clarified. In particular to allow proper data reproducibility the amount of root exudates (in terms of key compounds or total carbon for instance) must be reported and normalization of treatments among root exudates with respect to for instance total C must be performed. Otherwise we cannot appreciate if differences among root exudates may relate to different chemical composition (i.e. presence of elicitors) or to nutrient supplementation. This would be an additional proof of what authors later clarified with synthetic medium.

6. Line 301. A figure with this hypothesis could be appreciated, where authors suggest the level where the interaction between plant compounds and QS system may occur. An additional experiment with null mutants of the flavonoid receptor could allow to better define the molecular level of interaction (see line 375).

6. PLOS authors have the option to publish the peer review history of their article (what does this mean?). If published, this will include your full peer review and any attached files.

Reviewer #1: No

Reviewer #2: No

---

## [Author Response · Author response to Decision Letter 0]

6 Aug 2020

We have revised and appropriate changes have been made in the manuscript.

2.Thank you for stating the following in the Acknowledgments Section of your manuscript:

[LABV received a Fellowship 384814 from Consejo Nacional de Ciencia y Tecnología.]

[SB IN212920 from PAPIIT, DGAPA, UNAM

GTT PICT2016-0210 

GTT PIP 1220130100420CO CONICET

The funders had no role in study design, data collection and analysis, decision to publish, or preparation of the manuscript.]

We have deleted the funding from the Acknowledgments Section and modified the Funding Statement.

"This work was supported by grant IN212920 from PAPIIT, DGAPA, UNAM to SB and grants PICT2016-0210 and PIP 2014-0420 to GTT. LABV received a Fellowship 384814 from Consejo Nacional de Ciencia y Tecnología. There was no additional external funding received for this study. The funders had no role in study design, data collection and analysis, decision to publish, or preparation of the manuscript."

 

Point-to-point reply

Reviewer #1: This paper is an extension of very exciting previous work from the same group, which showed that conjugative transfer of pRetCFN42a is enhanced on plant surfaces and in nodules, using a very elegant labeling system. The current work seeks to address the nature of the effect of the plant on transfer. Different plants were examined (Corn, Alfalfa and Bean) and all promoted transfer, perhaps due to nutritional effects. Exudates from seeds/roots of alfalfa and bean were compared and there was a stronger effect of the host plant, bean. Known plant compounds were tested for stimulatory effects on conjugation, and an effect of two nodulation inducers, apigenin and naringenin, was noted. In all cases, transfer by conjugation required the AHL type QS system previously shown to be involved in transfer of this plasmid.

Overall this is a very well organized piece of work that adds to the story. It is very well written and easy to follow. I do not have too much to criticize except for minor editorial things (see below), but do have some suggestions for improving the paper or directing future work. Some of this might go in the discussion.

- Thank you, we have included many of the suggested ideas in the discussion.

1) It should probably be cited that conjugation of the plasmid pRL8JI (also a type I conjugation system) is stimulated by homoserine, a non-protein amino acid that is present in high concentration in pea root exudates (Vanderlinde et al. 2014, Environ Microbiol 16: 205-217).

Response: We have included this datum in the Introduction (lines 53, 82-83) and the Discussion (lines 354-356).

2) Some speculation as to other compounds that might be specific to beans or legumes in general would not be a bad idea. It is a little disappointing to look only at nod gene inducers.

Response: Yes, we had not mentioned specific compounds, but in the discussion, we mentioned in lines 356-360: “Our results showed that the sole presence of the plant plays an important role in increasing the conjugation frequency. This phenomenon could be due to secretion of signals that can be involved in regulation of CT, secretion of metabolites that the bacteria could use as energy supplies or as a support where the bacteria can adhere and conjugate”. Now we have re-enforced these ideas in the Discussion (lines 362-366).

3) Do the effects of naringenin and apigenin depend on the presence of a functional nodD gene ? This could be assayed fairly easily in a pSym cured derivative of CFN42, like your recipient, once it has acquired the p42a.

Response: We do not know if the effect is mediated directly by a functional nodD, but we have to point out that CFN42 has three copies of nodD. We think, with evidence provided by Ling et al (2016), that it could be a LysR-like transcriptional activator (actually, there are 69 in the genome of R. etli CFN42), including the nodD genes, but also other proteins. Also, Ling et al found that AhaR is a LysR-like transcriptional activator that detects flavonoids. By BLASTp we found some proteins similar to AhaR in CFN42. Of course, this question needs to be addressed in the future and the assay suggested, using a pSym- cured derivative of CFN42, could be achieved when we are able (hopefully soon) to return to work in the lab. We included an appropriate discussion of this issue in the discussion of the manuscript (Lines 387-398).

4) Ways of getting at what the particular compounds in exudates might be could be discussed. In this context the method used in Rosenblueth et al. (1998, MGG, 258:587) and Yost et al. (2004, Microbiology, 152:2061) could be mentioned. I.e. depletion of nutrients in exudates by plasmid cured strains to see what is left over and still has the ability to induce transfer.

Response: Yes, the reviewer is right. Those papers describe strategies to search which molecules induce certain genes (using transcriptional fusions) or which replicon is needed to employ certain sugars. We discuss about these strategies and will direct some future work (Lines 398-404) 

5) More could be made of the fact that genes encoding Xre-type regulators have been found in conjugation systems, including the one from pRetCFN42a, strongly suggesting additional effector molecules that have an input into regulation of transfer. It has always been a strong possibility that some plant produced compounds could be the effectors that interact with these regulators.

Response: We included this in the discussion (Lines 404-411). Regulators with Xre-domains (helix-turn-helix domain similar to that of the Lambda Cro and CI proteins) are regulators that have been described in pRet42a of R. etli CFN42 as RHE_PA00165 (Lopez-Fuentes et al., 2015), in pRleVF39b of R. leguminosarum VF39SM as trbR (Ding et al., 2013) and in ICEMlSymR7A of M. loti as qseC (Ramsay et al., 2013). RHE_PA00165 is needed by pRet42a to transfer from genomic backgrounds different from that of the wild-type. A mutation in trbR leads to a 1000-fold increase in pRleVF39b transfer. qseC modulates the excision and conjugative transfer of ICEMlSymR7A (Ramsay et al., 2013).

Many experts in the field don't like the use of the word "conjugal" and much prefer conjugative. Laura Frost, whom you cite in ref 1, gets quite angry at people who use conjugal.

Response: Corrected

Line 38. Homoserine is missing an e; same for line 64 and elsewhere

Response: Done

Line 49. Should be a semicolon (;) not a comma, after evaluated.

Response: Done

Line 195. no transconjugants (not not)

Response: Done

Lines 301/302. Not sure "above them in hierarchical order is correct. In agrobacterium, opines come first, with QS below opines in the cascade. But QS mutants (traR or traI will still not transfer in the presence of opines. Your situation could be the same. Maybe you need to explain what you mean by hierarchy here.

Response: Yes, the reviewer is right. We have not explained our idea properly. The sentence was changed to: “These results indicate that although some molecules present in plant exudates improve CT of pRet42a, the QS system is essential for plasmid transfer.” Lines 309-310

Table 2. Not all of these compounds are flavonoids, so the title is inaccurate. I suggest plant compounds, or plant produced phenolic compounds.

Response: Accepted. The title was changed to “Characteristics of the different plant compounds used”

References: some titles are written in sentence case (correct), others in title case (capital letters on most nouns and other important words) - be consistent. This is a common error caused by trusting reference manager programs and importing papers from PubMed etc. Also double check for accents on author names - some seem to be missing even on Spanish names, though this may be how the names were written on the original publication

Response: Done

 

Reviewer #2: The manuscript PONE-D-20-19518 reports data from experiments aimed to clarify the role of plant-derived compounds on the regulation of conjugative transfer of plasmid pRet42a from the legume microsymbiont Rhizobium etli. Obtained results indicate that quorum sensing system is playing the main role in the regulation, but also that compounds in plant root exudates may modulate the extent of conjugal transfer.

The results are interesting and novel, tough not conclusive over a mechanistic interpretation of the phenomenon. Consequently, the paper is mainly descriptive, though opening the way to several future experiments. A key experiment for a more mechanistic report could be the used of a nodD gene mutant strain (see point 6 below).

However, there are many points which need to be clarified and additional details reported to allow readers to better understand the results and let reproduce the experiments shown. Possible simple additional experiments could also be performed (see below).

Thank you. We addressed the nodD issue in point 6 (below). Responses to the additional details are mentioned in each case (see below).

1. Line 144. Please indicate the OD600 or the number of cells used.

Response: Done. Lines 138-141: “Conjugations between the strains were done biparentally, using overnight cultures grown to stationary phase, ca. 1.2 OD600nm [25]. Conjugations in liquid medium were done as follows: donor and recipient strains were mixed in a 1:1 volume ratio in 5 ml of PY or MM medium at a final OD600nm of 0.05 and incubated at 30 ºC overnight at 200 rpm.”

2. Line 146. Concentrations of used flavonoids should be indicated here.

Response: Done. Lines 141-142: “Flavonoids were included, respectively, at different concentrations (2 �M, 20 �M, and 50 �M).”

3. Lines 160 and followings. The plant assay must be described in detail. For instance, the volume used and the number of plants and root lengths. The number of planted cells (did you perform dilutions? Which volume of plant medium was taken?)

Response: The Material and methods section was rewritten and the requested details were included. 

Lines 142-148: “ Conjugations with exudates and extracts of roots and nodules were done in liquid medium by mixing 2.5 ml of Fahraeus containing the exudates or extracts (or nitrogen-free Fahraeus nutrient solution as control), 2.5 ml of PY medium, 0.25 ml of donor and 0.25 ml recipient (each ca. 1.2 OD600nm). The mixtures were incubated at 30 ºC overnight, and then centrifugated at 5000 rpm for 2 minutes and the supernatant was discarded. The pellet was suspended in 1 ml of 10 mM MgSO4 0.01% Tween 40 (vol/vol).”

Lines 158-174: “Seeds from P. vulgaris cv Negro Jamapa were sterilized and germinated as previously described by Bañuelos-Vazquez et al [32]. Seedlings of two dpg (3-4 cm in length) were introduced in tubes with 40 ml of nitrogen-free Fahraeus nutrient solution [38] and in the indicated experiments, also supplemented with a carbon (succinic acid 0.01M) and nitrogen source (ammonium chloride 0.01M) [38]. Tubes were inoculated with donor and recipient strains adjusted at a final 0.05 OD600nm, in a 1:1 ratio. The conjugation frequency was measured at 1, 10 and 20 dpi. For each experiment, samples of three plants from medium in presence or absence of plants were collected. The 40 ml of medium from the tubes were centrifugated at 5000 rpm, supernatant was discarded, and the pellet was resuspended in 1 ml of 10 mM MgSO4 0.01% Tween 40 (vol/vol). Serial dilutions were plated on the selective media supplemented with the corresponding antibiotics. Root samples were introduced in Falcon tubes with 30 ml of nitrogen-free Fahraeus nutrient solution and subjected to ultrasound for 20 min in a Branson 200 ultrasonic cleaner. Then, the roots were taken out and the medium was centrifuged for 15 min at 5000 rpm, at 4 ºC to recover the bacteria removed from the root's surface. The pellet was resuspended in 1 ml of 10 mM MgSO4 0.01% Tween 40 (vol/vol). Serial dilutions were plated on the selective media supplemented with the corresponding antibiotics. All experiments were repeated at least three times.”

4. Line 195-205. Why at at 20 dpi there were not transconjugants? This is really surprising since you observe transconjugants at 10 dpi. Were rhizobial cells titres comparable between 10 dpi and 20 dpi? Or rhizobial cells died after 10 dpi? This point strongly needs a clarification and hypotheses driven by possible additional control experiments, to support your sentence on line 352.

Response: Bacteria were plated 20 dpi, but when plants are not present, transconjugants did not grow. Donor and recipient titres were lower, indicating that a fraction of bacteria died. That is the reason that we suggest in (now line 361) “This result suggests that plants could also be important for long-term bacterial survival". 

We modified the following sentences to clarify:

Line 202-204: “It is worth to mention that comparison of the number of donor and recipient bacteria at 10 dpi and 20 dpi, showed that there were ca. 2-4 million less UFC at 20 dpi.” 

Line 207-210: “The lack of transconjugants at 20 dpi and the decrease in donor and recipient bacteria in the plant-free media allows us to consider the possibility that this may be caused by a nutrient limitation in the absence of plants”

Line 360-361: “Moreover, the lack of transconjugants (and the reduction of donor and recipient bacteria) at 20 dpi in absence of plants was remarkable”

5. Line 232. Experiment with plant root exudates must be clarified. In particular to allow proper data reproducibility the amount of root exudates (in terms of key compounds or total carbon for instance) must be reported and normalization of treatments among root exudates with respect to for instance total C must be performed. Otherwise we cannot appreciate if differences among root exudates may relate to different chemical composition (i.e. presence of elicitors) or to nutrient supplementation. This would be an additional proof of what authors later clarified with synthetic medium.

Response: As stated in response to query number 3, we modified Material and methods to provide the details and clarify the experiments. As the reviewer mentioned “we cannot appreciate if differences among root exudates may relate to different chemical composition or to nutrient supplementation”, we want to reinforce that exudates and extracts were only prepared in nitrogen-free Fahraeus nutrient solution. Thus, the differences observed among control and extracts or exudates, are due the presence of the molecules secreted by the plant, not by any nutrient supplementation.

We did not measure key compounds or total carbon in either exudates or extracts. Plant extracts and exudates could contain several hundreds of different compounds (Tawaraya et al., 2014, Wang et al., 2019) (we included some discussion about this in the discussion section). We think that not all the compounds will be used (or even detected) by the bacteria, thus, the amount of total carbon would not give much information and the chemical composition of the extracts and exudates are not the scope of the manuscript. Moreover, the molecules that regulate CT usually work at low concentrations, so the C contents could not reflect its potential as CT enhancer. Of course, it will be a future aim when we are able to go back to the laboratory, to identify the secreted molecules that are actually increasing the conjugative transfer.

6. Line 301. A figure with this hypothesis could be appreciated, where authors suggest the level where the interaction between plant compounds and QS system may occur. An additional experiment with null mutants of the flavonoid receptor could allow to better define the molecular level of interaction (see line 375).

Response: We do not know if the effect is mediated directly by a functional nodD, but we have to point out that CFN42 has three copies of nodD. We think, with evidence provided by Ling et al. (2016), it could be a LysR-like transcriptional activator (actually, there are 69 in the genome of R. etli CFN42: 46 in the chromosome, 6 in pRet42f, 8 in pRet42e, 5 in pRet42d -3 (nodD are included), 3 in pRet42c and 1 in pRet42b), this includes nodD genes, but also other proteins. We cannot disregard that the bioinformatics annotation could miss some others. Also, Ling et al. (2016) found that AhaR is a LysR-like transcriptional activator that detects flavonoids. By BLASTp we found some proteins similar to AhaR in CFN42. Thus, making a null mutant of flavonoid receptors would include at least a triple mutant and, nevertheless, many LysR-like genes would still be available in the genome. This question needs to be addressed in the future, when we are able to return to work in the lab. We have included an appropriate discussion of this issue in the manuscript and also a figure with our hypothesis (Lines 387-398).

Ding H, Yip CB & Hynes MF (2013) Genetic characterization of a novel rhizobial plasmid conjugation system in R. leguminosarum bv. viciae strain VF39SM. Journal of bacteriology 195: 328-339.

Ling J, Wang H, Wu P, Li T, Tang Y, Naseer N, Zheng H, Masson-Boivin C, Zhong Z & Zhu J (2016) Plant nodulation inducers enhance horizontal gene transfer of Azorhizobium caulinodans symbiosis island. Proceedings of the National Academy of Sciences of the United States of America 113: 13875-13880.

Lopez-Fuentes E, Torres-Tejerizo G, Cervantes L & Brom S (2015) Genes encoding conserved hypothetical proteins localized in the conjugative transfer region of plasmid pRet42a from Rhizobium etli CFN42 participate in modulating transfer and affect conjugation from different donors. Frontiers in microbiology 5: 793.

Ramsay JP, Major AS, Komarovsky VM, Sullivan JT, Dy RL, Hynes MF, Salmond GP & Ronson CW (2013) A widely conserved molecular switch controls quorum sensing and symbiosis island transfer in Mesorhizobium loti through expression of a novel antiactivator. Molecular microbiology 87: 1-13.

Tawaraya K, Horie R, Saito S, Wagatsuma T, Saito K & Oikawa A (2014) Metabolite Profiling of Root Exudates of Common Bean under Phosphorus Deficiency. Metabolites 4: 599-611.

Wang Y, Ren W, Li Y, Xu Y, Teng Y, Christie P & Luo Y (2019) Nontargeted metabolomic analysis to unravel the impact of di (2-ethylhexyl) phthalate stress on root exudates of alfalfa (Medicago sativa). The Science of the total environment 646: 212-219.

---

## [Decision Letter · Decision Letter 1]

13 Aug 2020

Role of plant compounds in the modulation of the conjugative transfer of pRet42a

PONE-D-20-19518R1

Dear Dr. %Torres-Tejerizo%,

We’re pleased to inform you that your manuscript has been judged scientifically suitable for publication and will be formally accepted for publication once it meets all outstanding technical requirements.

Kind regards,

Francisco Martinez-Abarca, Ph.D.

Academic Editor

PLOS ONE

Additional Editor Comments (optional):

Only a minor change isvrequired as pointed out by reviewr 2.

Reviewers' comments:

Reviewer's Responses to Questions

**Comments to the Author**

1. If the authors have adequately addressed your comments raised in a previous round of review and you feel that this manuscript is now acceptable for publication, you may indicate that here to bypass the “Comments to the Author” section, enter your conflict of interest statement in the “Confidential to Editor” section, and submit your "Accept" recommendation.

Reviewer #1: All comments have been addressed

Reviewer #2: All comments have been addressed

2. Is the manuscript technically sound, and do the data support the conclusions?

Reviewer #1: Yes

Reviewer #2: (No Response)

3. Has the statistical analysis been performed appropriately and rigorously? 

Reviewer #1: Yes

Reviewer #2: (No Response)

4. Have the authors made all data underlying the findings in their manuscript fully available?

Reviewer #1: Yes

Reviewer #2: (No Response)

5. Is the manuscript presented in an intelligible fashion and written in standard English?

Reviewer #1: Yes

Reviewer #2: (No Response)

6. Review Comments to the Author

Reviewer #1: The authors have addressed my minor concerns and provided a good response to my suggestions (and, in my view, those of the other reviewer). One very small issue that remains is that I don't think they quite grasped what I meant when I suggested using growth cured strains (or strains with mutations in given genes) to deplete the plant exudates and/or extracts of most the metaboliites. leaving behind only those that might induce transfer for the missing plasmid (this could indeed be assayed using fusions as they suggest). Such an approach would narrow down the number of compounds in exudates that might need to be screened. Not a big deal.

Reviewer #2: The manuscript has been revised by authors and amended following the suggestions from the previous reviewing assessment. I'm satified with the present version of the manuscript.

Minor editorial changes:

Line 221: Change UFC to CFU

7. PLOS authors have the option to publish the peer review history of their article (what does this mean?). If published, this will include your full peer review and any attached files.

Reviewer #1: No

Reviewer #2: No

---

## [Editor Report · Acceptance letter]

17 Aug 2020

PONE-D-20-19518R1 

Role of plant compounds in the modulation of the conjugative transfer of pRet42a 

Dear Dr. Torres Tejerizo:

I'm pleased to inform you that your manuscript has been deemed suitable for publication in PLOS ONE. Congratulations! Your manuscript is now with our production department. 

Kind regards, 

on behalf of

Dr. Francisco Martinez-Abarca 

Academic Editor

PLOS ONE